# Social Stress Increases Vulnerability to High-Fat Diet-Induced Insulin Resistance by Enhancing Neutrophil Elastase Activity in Adipose Tissue

**DOI:** 10.3390/cells9040996

**Published:** 2020-04-16

**Authors:** Shinichiro Motoyama, Hiroyuki Yamada, Keita Yamamoto, Noriyuki Wakana, Kensuke Terada, Masakazu Kikai, Naotoshi Wada, Makoto Saburi, Takeshi Sugimoto, Hiroshi Kubota, Daisuke Miyawaki, Daisuke Kami, Takehiro Ogata, Masakazu Ibi, Chihiro Yabe-Nishimura, Satoaki Matoba

**Affiliations:** 1Department of Cardiovascular Medicine, Graduate School of Medical Science, Kyoto Prefectural University of Medicine, Kyoto 602-8566, Japan; 2Department of Regenerative Medicine, Graduate School of Medical Science, Kyoto Prefectural University of Medicine, Kyoto 602-8566, Japan; 3Department of Pathology and Cell Regulation, Graduate School of Medical Science, Kyoto Prefectural University of Medicine, Kyoto 602-8566, Japan; 4Department of Pharmacology, Graduate School of Medical Science, Kyoto Prefectural University of Medicine, Kyoto 602-8566, Japan

**Keywords:** social stress, insulin resistance, neutrophil, neutrophil elastase, adipose tissue, heat-shock protein 72

## Abstract

Social stress (SS) has been linked to the development of cardiovascular disease (CVD), which is closely associated with insulin resistance (IR); however, the causal effect of SS on IR remains unclear. The 8-week-old male C57BL/6 mice were exposed to SS by housing with a larger CD-1 mouse in a shared home cage without physical contact for 10 consecutive days followed by high-fat diet (HFD) feeding. Control mice were housed in the same cage without a CD-1 mouse. After 6 weeks of HFD, insulin sensitivity was significantly impaired in stressed mice. While the percentage of classically activated macrophages in epididymal white adipose tissue (eWAT) was equivalent between the two groups, the percentage of lymphocyte antigen 6 complex locus G6D (Ly-6G)/neutrophil elastase (NE)-double positive cells markedly increased in stressed mice, accompanied by augmented NE activity assessed by ex vivo eWAT fluorescent imaging. Treatment with an NE inhibitor completely abrogated the insulin sensitivity impairment of stressed mice. In vitro NE release upon stimulation with a formyl peptide receptor 1 agonist was significantly higher in bone marrow neutrophils of stressed mice. Our findings show that SS-exposed mice are susceptible to the development of HFD-induced IR accompanied by augmented NE activity. Modulation of neutrophil function may represent a potential therapeutic target for SS-associated IR.

## 1. Introduction

Social stress (SS) has been associated with the development of cardiovascular disease (CVD) [1,2]. SS is also associated with the development of metabolic syndrome and type 2 diabetes mellitus (T2DM) [3,4], in which insulin resistance (IR) plays a critical role [5,6]. Considering that IR is causally implicated in the pathogenesis of atherosclerosis [7], and patients with IR are more likely to develop CVD than healthy individuals [8], SS-induced IR is thought to be a key contributor to the development of SS-related CVD; however, this precise mechanism has not been fully investigated.

Previous studies have investigated the effect of SS on adipose tissue inflammation as a critical contributor to the development of IR [9,10]; however, cellular and molecular mechanisms of SS-induced IR remain elusive. Exposure to SS leads to the activation of the hypothalamic–pituitary–adrenal (HPA) axis and sympathetic nervous system (SNS) [11,12]. Activation of the SNS has been shown to not only impact hemodynamic status but also to modulate bone marrow (BM) homeostasis via β3-adrenoreceptor activation, leading to enhanced mobilization of myeloid cells in the peripheral circulation [13,14]. Consequently, the number of peripheral blood neutrophils is markedly increased after chronic stress burden [15]. Furthermore, recent studies showed that activation of neutrophil elastase (NE) was involved in the development of IR through impairment of insulin signaling [16,17]. These findings led us to investigate whether SS-related alteration of BM homeostasis could accelerate the development of high-fat diet (HFD)-induced IR.

In the current study, we show that socially stressed mice are susceptible to the development of HFD-induced IR, accompanied by augmented NE activation in epididymal white adipose tissue (eWAT). Treatment with an NE inhibitor completely abrogated the effect of SS on IR development. Our findings suggest that modulation of NE activity could be a potential therapeutic strategy for preventing SS-related IR.

## 2. Materials and Methods

### 2.1. Experimental Animals

All experiments were performed with strict adherence to “Directive 2010/63/EU” of the European Parliament and to the Guidelines for Animal Experiments of the Kyoto Prefectural University of Medicine, following approval by the Institutional Animal Care and Use Committee of the Kyoto Prefectural University of Medicine.

Male wild-type mice (C57BL/6) and male CD-1 mice were obtained from Shimizu Laboratory Supplies Co., Ltd. (Kyoto, Japan). The 8–10-week-old wild-type mice were exposed to SS according to the protocol reported by Golden et al. [18] with modifications as described below. After screening of aggressor CD-1 mice, each CD-1 resident mouse received a wild-type intruder mouse and the two animals were separated by a perforated partition, which allowed for continuous visual, auditory, and olfactory contact with no physical interaction. The partition was solid and tightly fixed in the bottom of the cage and the size of perforation was small and located 3 cm above the bottom. Further, during the 10 days of SS, this perforated partition was not removed. Thus, it was unlikely to share the food pellets and feces between compartments inside the same cage (Appendix A). During the 10 days of SS, the intruder wild-type mouse was exposed daily to a novel resident’s home cage to prevent any habituation to the resident aggressor. Control mice were individually housed in the same cage without CD-1 mice. After behavior analysis, mice were housed in groups (three to five animals per cage) and fed a HFD (energy content: 62% fat, 18.2% protein, and 19.6% carbohydrate; Oriental Yeast Co., Tokyo, Japan) for 6 weeks. For NE inhibitor treatments, NE inhibitor (GW311616A, AdipoGen^®^, San Diego, CA, USA) was administered by oral gavage every other day (2 mg/kg/day) during 6 weeks of HFD feeding, while control mice were given the vehicle (distilled water) in the same manner, as previously described [17]. Animals were housed in a room maintained at 22 °C under a 12 h light/dark cycle and provided with drinking water ad libitum. After 6 weeks of HFD feeding, mice were euthanized by transcardial perfusion under anesthesia induced by isoflurane (2%; 0.2 mL/min). Blood samples were obtained in the morning (08:00–10:00) after an overnight fast. The scheme of the design of the different experiments was shown in Appendix A.

### 2.2. Assessment of Body Weight and Food Intake

Body weight was measured for stressed and control mice biweekly after starting HFD feeding. Average food intake per cage of stressed and control groups was monitored during HFD feeding, including spillage, by measuring the weight of food pellets (g) in cages with 3 similarly aged mice. Cumulative caloric intake over 6 weeks was calculated by multiplying the total weight of food intake by total calorie of food pellets (5.062 kcal/g).

### 2.3. Behavior Analysis

Before HFD feeding, two behavior analysis tests were performed, as previously described [19,20]. The social interaction test was used to examine social approach toward an unfamiliar mouse and time spent in the interaction zone, when the target was absent or present; this was measured using a charge-coupled device (CCD) video camera (JIN-608AC; Kyohritsu Electronic Industry Co., Ltd., Osaka, Japan). The social interaction ratio was obtained by dividing interaction time spent in the presence of the target by time spent in the absence of the target. In the sucrose preference test, mice were given a choice between two 50 mL bottles containing either 1% sucrose or water. To control for side preference, the position of the bottles was switched daily. The tubes were weighed daily to determine intake. Sucrose preference data were calculated as shown in Equation (1): (1)$$\%\;\mathrm{Sucrose}\;\mathrm{intake}=\frac{\mathrm{Sucrose}\;\mathrm{Cosnumed}\;\left(g\right)}{\mathrm{Water}\;\mathrm{Consumed}\;\left(g\right)+\mathrm{Sucrose}\;\mathrm{Consumed}\;\left(g\right)}$$

### 2.4. Glucose Tolerance Test (GTT) and Insulin Tolerance Test (ITT)

GTT was performed on 16-week-old mice after an overnight fast. Blood glucose concentrations were measured prior to and 15, 30, 60, 90, and 120 min after intraperitoneal injection of glucose (2 g/kg body weight). For ITT, insulin (1 U/kg body weight in 0.1% distilled water; Humulin R-Insulin, Eli Lilly Japan K.K. Kobe, Japan) was intraperitoneally injected after an overnight fast. Blood glucose concentrations were measured prior to and 30, 60, 90, and 120 min after injection.

### 2.5. Enzyme-Linked Immunosorbent Assay (ELISA)

Blood was collected into tubes from the left ventricle. Blood samples were obtained in the morning (08:00–10:00) after an overnight fast. Serum levels of noradrenaline, corticosterone, and HSP72 were examined in 10-week-old mice before HFD feeding and serum insulin level was examined in 16-week-old mice after 6 weeks of HFD feeding. Serum was separated by centrifugation at 2000× *g* for 20 min and stored at −80 °C. Serum levels were estimated using ELISA kits (BA E-5200; Labor Diagnostika Nord, Nordhorn, Germany; ab108821; Abcam plc, Cambridge, UK; Mouse insulin ELISA KIT, MS303; Morinaga Institute of Biological Science, Yokohama, Japan; MBS9364287; MyBioSource, San Diego, CA, USA) according to the manufacturer’s instructions.

### 2.6. Immunohistochemistry

Epididymal adipose tissue was removed immediately after saline perfusion and embedded in paraffin. For the immunological staining of lymphocyte antigen 6 complex locus G6D (Ly-6G), Alexa Fluor 647-conjugated anti-mouse Ly-6G (clone 1A8; BioLegend, San Diego, CA, USA) was used. For Mac2, anti-mouse Mac2 antibody (Clone M3/38; Cedarlane, Burlington, Ontario, Canada) and Alexa Flour 555-conjugated secondary antibody (Thermo Fisher Scientific, Waltham, MA, USA) were used. For NE, anti-mouse-neutrophil elastase antibody (Abcam plc, Cambridge, UK) and Alexa Fluor 555-conjugated secondary antibodies (Thermo Fisher Scientific) were used. Nuclei were labeled using 4’,6-diamidino-2-phenylindole (DAPI) (62248; Thermo Fisher Scientific), and sections were examined using an LSM 510 META confocal microscope (Carl Zeiss, Jena, Germany). For negative control, non-immune immunoglobulin and Alexa Fluor 555-conjugated secondary antibodies (Thermo Fisher Scientific) were used. Positive staining was evaluated using Image J v1.48 software (https://imagej.nih.gov/ij/index.html). The percentages of Ly-6G- and Ly-6G/NE-positive stained nuclei in total number of crown-like structures were assessed per 3 sections from 8–10 animals from each group. The percentage of Mac2-stained nuclei in total nuclei number of crown-like structures was assessed per 3 sections from 5 animals from each group. A total of 4 to 6 representative images were chosen from 3 sections at random in each animal, and then analyzed.

### 2.7. Blood Leukocyte Counts

Blood samples were obtained in the morning (08:00–10:00) after an overnight fast in 10-week-old mice before HFD feeding and in 16-week-old mice after 6 weeks of HFD feeding. Blood was harvested from the left ventricle, collected in EDTA or heparin, and analyzed using the ADVIA 120 hematology system (Siemens Healthcare, Erlangen, Germany).

### 2.8. Real-Time Polymerase Chain Reaction (RT-PCR)

Total RNA was extracted from adipose tissue and liver using the RNeasy Lipid Tissue Mini Kit (74804; Qiagen, Hilden, Germany), and reverse transcribed to prepare cDNA, using the TAKARA Prime Script RT reagent Kit with gDNA Eraser (RR047A; Takara Bio, Shiga, Japan). Real-time PCR was performed using a Thermal Cycler Dice system (Takara Bio), with the KAPA SYBR^®^ FAST Universal qPCR Kit (KK4602; KAPA Biosystems, Wilmington, MA, USA). Dissociation curves were examined for aberrant formation of primer dimers. Threshold cycle (CT) values were normalized to GAPDH, and relative expression was calculated by the ΔΔCT method. Data were expressed as gene expression levels relative to those of controls. The following primers were used: NE: forward, 5’-CCTTGGCAGACTATCCAGCC-3’; reverse, 5’-GACATGACGAAGTTCCTGGCA-3’; TNF-α: forward, 5’-TCCCAGGTTCTCTTCAAGGGA-3’; reverse, 5’-GGTGAGGAGCACGTAGTCGG-3’; MCP-1: forward, 5’-GGCTCAGCCAGATGCAGTTAA-3’; reverse, 5’-CCTACTCATTGGGATCATCTTGCT-3’; ICAM-1: forward, 5’-AGCACCTCCCCACCTACTTT-3’; reverse, 5’-AGCTTGCACGACCCTTCTAA-3’; IL-1β: forward, 5’-AGAGCCCATCCTCTGTGACTCA-3’; reverse, 5’-TCATATGGGTCCGACAGCACGA-3’; IL-6: forward, 5’-ACAACCACGGCCTTCCCTACTT-3’; reverse, 5’-CACGATTTCCCAGAGAACATGTG-3’; MIP-2: forward, 5’-CCAACCACCAGGCTACAGG-3’; reverse, 5’-GCGTCACACTCAAGCTCTG-3’; GAPDH: forward, 5’- TGTCCGTCGTGGATCTGAC-3’; reverse, 5’-CCTGCTTCACCACCTTCTTG-3’.

### 2.9. Flow Cytometry and Cell Sorting (FACS)

Peripheral blood monocytes and neutrophils, as well as epididymal white adipose tissue (eWAT) macrophage and neutrophils, were analyzed using FACS analysis.

For staining of peripheral monocytes, anti-mouse FITC-conjugated B220 (clone RA3-6B2; BD Biosciences, Franklin Lakes, NJ, USA), CD11c (clone HL; BD Biosciences), NK1.1 (clone PK136; BD Biosciences), CD49b (clone DX-5; BD Biosciences), CD90.2 (clone 53-2.1; BD Biosciences), Ly-6G (clone 1A8; BD Biosciences), F4/80 (clone BM8; BioLegend), and I-Ab (clone 25-9-17, BioLegend) antibodies were used as lineage markers. Blood cells were stained with APC-conjugated CD11b (clone M1/70; BD Biosciences) and APC-Cy7-conjugated Ly-6C (clone HK1.4; Biolegend) antibodies, as previously described [21].

For staining peripheral blood neutrophils, anti-mouse FITC-conjugated B220 (clone RA3-6B2; BD Biosciences), NK1.1 (clone PK136; BD Biosciences), CD49b (clone DX-5; BD Biosciences), CD90.2 (clone 53-2.1; BD Biosciences), Ter119 (clone Ter119; BioLegend), and APC-Cy7-conjugated CD115 (clone AFS98; BioLegend) antibodies were used as lineage markers. Blood cells were stained with APC-conjugated CD45 (clone 30-F11; BioLegend), PerCP-Cy5.5-conjugated CD11b (clone M1/70; BD Biosciences), and PE-conjugated Ly-6G (clone 1A8; BD Biosciences) antibodies, as previously described [15].

For staining of eWAT macrophages, stromal vascular cells were stained with PerCP-Cy5.5-conjugated anti-CD45 (clone 30-F11; BD Biosciences), PE-conjugated anti-F4/80 (clone BM8; eBioscience, Wien, Austria), FITC-conjugated anti-CD11b (clone M1/70; BD Biosciences), PE-Cy7-conjugated anti-CD11c (clone N418; eBioscience), and APC-conjugated anti-CD206 (Clone C068C2, BioLegend) antibodies, which identify discrete M1 and M2 adipose tissue macrophage subsets in both lean and obese mice [22].

For staining eWAT neutrophils, stromal vascular cells were stained with PE-conjugated anti-CD11c (clone HL3; BD Biosciences), anti-F4/80 (clone T45-2342; BD Biosciences), FITC-conjugated anti-Ly-6G (clone 1A8; BD Biosciences), and APC-conjugated anti-CD11b (clone M1/70; BD Biosciences) antibodies [17]. Cells were sorted using an SH800 cell sorter (Sony, Tokyo, Japan).

### 2.10. Ex Vivo Neutrophil Elastase (NE) Activity

On experiment day, all animals were administered with 100 μL neutrophil elastase 680 FAST (Perkin Elmer, Boston, MA), via tail vein injection. The 680 FAST comprises two near-infrared (NIR) fluorochromes (VivoTag-S680; PerkinElmer) linked to the N- and C-terminus of the peptide PMAVVQSVP, a highly NE-selective sequence. NE680 FAST is optically silent in its native state and becomes highly fluorescent upon cleavage by NE and can therefore be used as a direct sensing agent for NE activity. After 5 h post-injection, the liver, eWAT, and lower limbs were harvested and then ex vivo imaging was immediately performed using an IVIS Lumina Series III optical imaging platform using the red filter sets (excitation range, 640 nm; emission, 670 nm longpass). We spectrally unmixed fluorescent data to obtain a signal specific to NE680 FAST.

We manually drew regions of interest (ROIs) encompassing the whole organs and the resulting signal was computed in the units of scaled counts per second. We carefully ensured that the size of the ROIs drawn across animal samples was consistent [16].

### 2.11. In Vitro Activation of Bone Marrow (BM) Neutrophils Upon Stimulation with Formyl Peptide Receptor (FPR) 1 Agonist

BM neutrophils (Lin^−^CD45^+^CD11b^+^Ly-6G^+^) were isolated from stressed and control mice before high-fat diet (HFD) feeding by flow cytometry. B220, NK1.1, CD49b, CD90.2, Ter119, and CD115 antibodies were used as lineage markers, as previously described [15]. The cells were resuspended in RPMI 1640 medium at 1 × 10^6^ cells/mL and incubated with or without the FPR 1-specific peptide agonist (WKYMVm; 400 nM; Sigma-Aldrich, St. Louis, MO, USA) for 30 min at 37 °C, as previously described [23]. The samples were centrifuged at 200× *g* for 10 min at 4 °C and then the supernatants and cell lysates were collected for measuring NE levels. NE concentrations were estimated using an ELISA kit (MBS009052; MyBioSource) according to the manufacturer’s instructions.

### 2.12. Statistical Analysis

We performed a Kolmogorov–Smirnov test for the normality of all continuous variables. A *p*-value above 0.05 indicated that data were normally distributed, and data were expressed as the mean ± standard error of the mean (SEM). Mean values were compared using analysis of variance (ANOVA) followed by a Tukey–Kramer test to analyze significant differences between the groups. Significant differences among groups for dependent variables were detected using two-way ANOVA: SS (Control versus Stress) and diet (before HFD feeding versus after HFD feeding). A *p*-value below 0.05 was considered statistically significant.

## 3. Results

### 3.1. SS Increases the Vulnerability to the Development of HFD-Induced IR

Before HFD feeding, stressed mice showed significantly higher concentrations of serum noradrenaline than control mice, while serum corticosterone levels were comparable between the two groups (Figure 1A). The social interaction test and the sucrose preference test did not show any differences between the two groups (Figure 1B), suggesting that SS induced in this experiment activates the SNS without evoking depression-like behavior. Glucose tolerance was significantly impaired in HFD-fed mice compared with in mice before HFD feeding; however, there was no difference between the two groups of control and stressed mice. In contrast, insulin sensitivity in stressed mice was significantly impaired after HFD feeding, resulting in the significantly higher AUC than that in HFD-fed control mice (Figure 1C). Consistently, serum insulin levels and homeostasis model assessment (HOMA)-IR after HFD feeding were significantly higher in stressed mice than in control mice (Figure 1D). These findings indicate that IR development was established in stressed mice after 6 weeks of HFD feeding. Mean body weight (BW) and cumulative caloric intake did not differ between the two groups (Appendix A). However, taking into consideration that average food intake per cage may mask the individual variations responsible for their stress levels, this finding needs to be interpreted with caution. The eWAT weight and eWAT weight/BW in stressed mice tended to be lower than those in control mice, but not significant (*p* < 0.09, *p* < 0.07 vs. control, respectively). Actual difference in serum noradrenaline level was very narrow, which was considered as a possible explanation for why eWAT weight and eWAT weight/BW did not reach statistical difference.

Although we did not examine the effects of longer periods of SS on insulin resistance in the lean mice, we examined the effect of repeated social defeat (RSD) that allowed for direct physical contact for 10 min each day during the 10 days of SS, leading to the development of depression-like behavior [18]. Social interaction ratio of RSD-exposed mice was significantly lower than those of control and SS-exposed mice (0.77 ± 0.11, *p* < 0.05); however, there was no difference in glucose and insulin tolerance between the three groups (Appendix A), suggesting that IR is not likely to be evoked in the lean mice despite varying amount of SS.

These findings show that SS-exposed mice are susceptible to the development of HFD-induced IR, which is independent of food intake and BW gain.

### 3.2. SS Augments Neutrophil Accumulation in eWAT

Epididymal adipocyte size was significantly increased after HFD feeding in both groups of mice; however, the difference in size was modest between the HFD-fed mice (Figure 2A,B). In contrast, the number of crown-like structures (CLSs) was markedly increased in HFD-fed stressed mice compared with HFD-fed control mice (Figure 2B), suggesting that augmented accumulation of inflammatory cells may contribute to the early onset of IR development in stressed mice. We therefore examined the fractions of peripheral blood monocytes and eWAT macrophages polarization [24]; unexpectedly, however, there was no difference between the two groups (Appendix A). Consistent with this, MCP-1 mRNA expression in eWAT and accumulated Mac-2-positive cells in CLSs were comparable between the two groups (Figure 2C, Appendix A). However, the percentage of Ly-6G-positive cells in CLSs was markedly higher in stressed mice than in control mice (Figure 2D). Given that neutrophils play a crucial role in the early stage of HFD-induced IR [16,17], we focused on the role of neutrophils in SS-induced IR. 

### 3.3. NE Activation Is Augmented in eWAT of Stressed Mice

To examine the role of neutrophils in eWAT inflammation, we performed immunohistological staining for NE. The percentage of Ly-6G/NE-double positive cells was markedly increased in stressed mice (Figure 3A). We performed RT-PCR analysis of eWAT and liver before HFD feeding as shown in Figure 3B. NE mRNA expression levels in eWAT were comparable between the two groups before HFD feeding; however, they were significantly increased in the eWAT of stressed mice after HFD feeding, leading to a significant enhancement compared with those in HFD-fed control mice. NE mRNA expression levels in liver were comparable before and after HFD feeding in both groups of mice, suggesting the possibility that neutrophil accumulation before HFD feeding was not affected by SS and that enhanced neutrophil accumulation and subsequent NE activation were exhibited after HFD feeding in stressed mice. Correspondingly, ex vivo NE activity assessed by fluorescent imaging was significantly augmented in eWAT of stressed mice, while those of the liver and lower limbs were comparable between the two groups (Figure 3C, Appendix A). These findings suggest that augmented NE activation in eWAT is likely responsible for the accelerated HFD-induced IR development in stressed mice.

### 3.4. NE Inhibitor Treatment Completely Abolishes the Accelerated IR Development in Stressed Mice

We next examined the effect of an NE inhibitor on SS-induced IR. Serum insulin levels and HOMA-IR were comparable between the NE inhibitor-treated stressed and control mice (Figure 4A). Likewise, glucose and insulin tolerance tests did not show any difference between the two groups after NE inhibitor treatment (Figure 4B). The percentage of Ly-6G-positive cells in CLSs was also comparable between the two groups after NE inhibitor treatment (Figure 4C). These findings support the notion that enhanced NE activation in eWAT is closely related to the augmented IR development in stressed mice.

### 3.5. In Vitro NE Release Is Augmented in BM Neutrophils of Stressed Mice

Although the percentage of Ly-6G/NE-double positive cells in CLSs was markedly increased in stressed mice, the number of peripheral blood neutrophils and the fraction of neutrophils in eWAT were comparable between the stressed and control mice (Appendix A). This finding led us to hypothesize that degranulation of NE is more augmented in stressed mice. Indeed, the concentration of NE in the culture supernatant upon stimulation with the formyl peptide receptor (FPR) 1-specific agonist was significantly higher in BM neutrophils of stressed mice than of control mice (Figure 5A), whereas the amount of NE before stimulation was comparable between the two groups (Figure 5B), suggesting that augmented NE release upon stimulation in eWAT contributes to the enhanced NE activity in stressed mice. To further examine the mechanism of enhanced NE degranulation, we examined the mRNA expression levels of tumor necrosis factor (TNF)-α, murine interleukin (IL)-8 homologue, and macrophage inflammatory protein (MIP) 2, because these cytokines have been reported to be involved in neutrophils priming to promote the release of NE [24]. mRNA expression levels of TNF-α and MIP2 were significantly higher in peripheral blood neutrophils of stressed mice than of control mice, while NE mRNA expression levels were comparable between the two groups (Figure 5C). We further examined the serum concentration of heat-shock protein (HSP) 72, which has been reported to be elevated in stressed mice via adrenoreceptor activation [25] and to increase the production of TNF-α and IL-8 in human BM neutrophils [26]. Serum HSP72 levels were significantly higher in stressed mice than in control mice (Figure 5D), suggesting that HSP72 plays a crucial role as a mediator of sterile inflammation, inducing the stress-evoked enhancement of NE activity.

## 4. Discussion

In this study, we showed for the first time that SS increases the vulnerability to HFD-induced IR and is accompanied by augmented NE activity in eWAT. Treatment with an NE inhibitor completely abrogated HFD-induced IR in stressed mice, suggesting that NE activity plays a crucial role in enhancing HFD-induced IR resulted from chronic stress exposure. Furthermore, in vitro NE release from BM neutrophils upon stimulation with FPR1 agonist significantly increased in stressed mice, whereas NE levels before stimulation were the same as in control mice. Our findings provide new insights into the causal association between stress burden and development of HFD-induced-IR, in which stress-evoked NE activity in eWAT may play a crucial role.

The causal effect of stress exposure on glucose homeostasis has been investigated in various animal models [9,10]. As shown by Uchida et al. [9], two weeks of restraint stress impaired insulin sensitivity in normal diet-fed mice and was accompanied by the augmented accumulation of macrophages and subsequent inflammatory response in adipose tissue. They also showed that BW and adipocyte size were significantly reduced in stressed mice versus control mice; however, these findings are not likely the case with the pathogenesis of IR in the clinical setting because BW and adipocyte size in metabolic syndrome usually increase, accompanied by the increase in adipose tissue weight [5,6]. In our experiment, stressed mice did not develop IR during normal diet feeding; however, they exhibited IR after HFD feeding along with an increase in BW and adipocyte size. This finding suggests the notion that moderate stress exposure, without directly affecting glucose homeostasis, can exert synergetic effects on the development of IR in combination with HFD feeding. Recently, Liu et al. [10] investigated the effects of chronic noise exposure on IR in HFD-fed mice and found that insulin sensitivity tended to be decreased in 8-week-old HFD-fed stressed mice; however, this was not significantly different to insulin sensitivity in HFD-fed control mice. This may be attributed to the difference in the intensity and duration of stress exposure due to the different experimental protocols. We did not examine the effect of SS on the late phase of IR development. The first infiltrating inflammatory cells after starting the HFD are neutrophils, but not macrophages, and we therefore focused on the early onset of HFD-induced IR to examine the contribution of SS-induced activation of neutrophils. Talukdar et al. showed that augmented accumulation of neutrophils in a few days after HFD feeding was still observed up to 90 days of HFD feeding [16]. Because the neutrophils are involved in the recruiting and activating ATMs during high-fat feeding, the effect of SS-induced activation of neutrophils on the late phase of IR development needs to be investigated in future studies.

The underlying mechanism of augmented eWAT NE activity in stressed mice may be attributed to a combination of various factors modulating neutrophil kinetics and function. Immunohistochemical staining showed that the percentage of Ly-6G-positive cells in CLSs was markedly higher in stressed mice than in control mice. However, the percentage fraction of neutrophils in eWAT assessed by flow cytometry was comparable between the two groups, which could be affected by the total numbers of leukocytes, endothelial cells, and stromal cells existing in the vascular stromal fraction (VSF). Indeed, the percentage fraction of eWAT macrophages, which constituted more than 10% of the VSF, increased slightly in stressed mice. In contrast, eWAT neutrophils constituted at most 1% VSF even after HFD feeding, making it difficult to detect minor differences in neutrophils fractions. Nevertheless, considering that Ly-6G-positive cells were scarcely detected in eWAT other than in regions of CLSs, the difference in the total number of neutrophils does not seem to be sufficient for exerting the augmented NE activity in stressed mice. We therefore examined the endogenous amount and release of NE upon stimulation in BM neutrophils. While the NE amount in BM neutrophils was equivalent between the stressed and control mice, NE release upon stimulation with FPR1 agonist slightly, but significantly, increased in BM neutrophils of stressed mice before HFD feeding. These findings suggest that stress-evoked neutrophil priming before HFD feeding is associated with the enhanced degranulation response to opsonized particles during HFD feeding, thereby contributing to the augmentation of NE activity in HFD-fed stressed mice.

In addition to the SNS and HPA axis responses to stress exposure [16,17], sterile inflammatory processes have emerged as key factors linking SS and inappropriate inflammation response [27,28]. Among various kinds of damage-associated molecular patterns (DAMPs), we focused on HSP72 as it is evoked by stress exposure via SNS activation [25], and was found to increase mRNA expression levels of *TNF-α* and *IL-8* in human BM neutrophils [26], which were elevated in the bronchoalveolar lavage in cystic fibrosis (CF) patients and directly contributed to the enhanced NE release [24]. Consistent with these findings, serum concentrations of HSP72 were significantly increased in stressed mice and mRNA expression levels of TNF-α and MIP2 (murine IL-8 homologue) in peripheral blood neutrophils were markedly elevated. In contrast, a protective effect of HSP72 on obesity-induced IR has also been reported [29,30]. Chung et al. [29] showed that HSP72 expression in skeletal muscle of obese, IR patients was decreased along with the impaired insulin signaling. Henstridge et al. [30] reported that overexpression of HSP72 in mouse skeletal muscle can prevent HFD-induced IR. While intracellular HSP72 in skeletal muscle exerts anti-inflammatory effects, extracellular HSP72 exhibits the opposite effect [31]. Indeed, serum concentration of HSP72 was significantly increased in patients with T2DM and was significantly correlated with IR [32]. These findings support the notion that extracellular HSP72, but not intracellular HSP72, is crucially involved in the pathogenesis of IR, at least in part, through the modulation of kinetics and function of innate immune cells.

Elgazar-Carmon et al. reported that neutrophils transiently infiltrated the parenchyma of intra-abdominal adipose tissue of HFD-fed mice and also observed a physical binding between neutrophils and adipocytes using immunohistochemistry analysis [33]. Consistent with in vivo findings, they demonstrated that adherence of mouse peritoneal neutrophils to a monolayer of 3T3-L1 mouse adipocytes was significantly exaggerated after stimulation of neutrophils using IL-8 analogue CXCL1 chemokine. They also showed that the adherence of neutrophils was prevented by preincubating with anti-ICAM-1 antibodies. Furthermore, Brake et al. reported that ICAM-1 mRNA expression was significantly increased in the abdominal fat of HFD-fed mice in a tissue-specific manner [34]. Considering that ICAM-1 expression in eWAT was comparable between stressed and control mice (Appendix A), and that CXCL2/MIP2 (murine IL-8 homologue) expression was significantly elevated in peripheral blood neutrophils of stressed mice, SS-induced priming of neutrophils is likely to contribute to the eWAT-specific enhancement of NE activation in stressed mice.

Mechanistic links between NE activation in adipose tissue and systemic insulin resistance remain obscure in this study. Walsh et al. reported that activated NE functioned as a Toll-like receptor (TLR) 4 activator in human bronchial epithelium [35]. Likewise, Talukdar et al. also showed that stimulation of intraperitoneal macrophages upon recombinant mouse NE significantly increased TNF-α mRNA expression in a TLR4-dependent manner [16]. We observed that TNF-α mRNA expression in eWAT was significantly increased in stressed mice after HFD feeding—the extent of which was 3-fold higher than that in HFD-fed control mice (Appendix A). Considering that the eWAT macrophage fraction was comparable between the two groups, augmented NE activation in neutrophils is likely to contribute to an increase in TNF-α mRNA expression in eWAT via activation of adipose tissue macrophages (ATMs). Accumulation of ATMs and subsequent inflammatory response have emerged as a critical contributor to promote whole-body insulin resistance [36,37]. TNF-α is one of the most commonly implicated cytokines in IR development and has been shown to inhibit the effect of insulin to suppress lipolysis in adipocytes [38,39]. Augmented release of lipolytic substrates such as non-esterified fatty acids (NEFAs) is critical for insulin resistance by impairing insulin-stimulated glucose uptake [40,41]. However, the relative contributions of NEFAs to skeletal muscle IR are less elucidated than hepatic IR [42], and therefore, integrated mechanisms linking between lipolysis in adipose tissue and stress-associated whole-body IR needs to be investigated in future studies.

The results of our study show that SS activates the SNS and subsequently increases the vulnerability to HFD-induced IR, accompanied by a marked increase in NE activity in eWAT. We also showed that treatment with an NE inhibitor completely abolished HFD-induced IR in stressed mice. Furthermore, in vitro NE activation of BM neutrophils of stressed mice was significantly augmented and accompanied by the increased mRNA expression of TNF-α and MIP2 in peripheral neutrophils. These findings support the notion that NE activation plays a critical role in stress-related IR development and shed new insights into the underlying mechanism of stress-related CVD development by focusing on the modulation of neutrophil function. As well as the pharmacological inhibition of NE, mice genetically deficient in NE exhibited more insulin sensitivity than wild-type mice after HFD feeding [16]. Consistently, α-1 anti-trypsin transgenic mice were resistant to HFD-induced IR [17]; however, few studies have focused on the association between stress response and HFD-induced IR. To elucidate the causal effect of stress-mediated neutrophil NE activation on HFD-induced IR, future studies using neutrophil-specific conditional knockout mice are needed.

## Figures and Tables

**Figure 1 cells-09-00996-f001:**
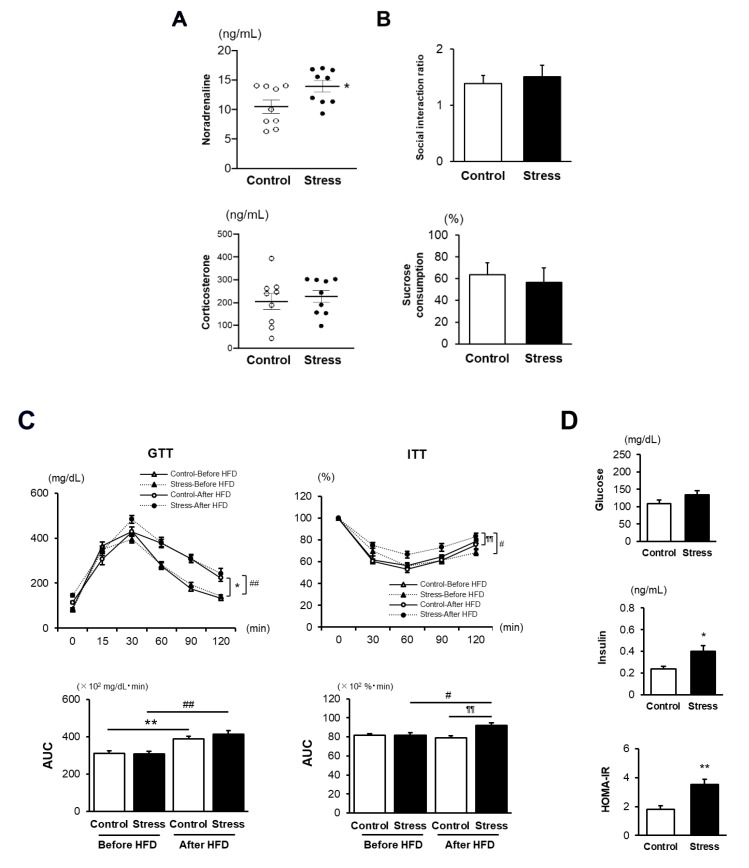
Social stress (SS) increases the susceptibility to the development of high-fat diet (HFD)-induced insulin resistance (IR). (**A**) Serum noradrenaline and corticosterone levels after SS. Values represent the mean ± SEM for nine (Control) and nine (Stress) mice. * *p* < 0.05 vs. Control. (**B**) Social interaction ratio and sucrose consumption after SS. Values represent the mean ± SEM for 10 (Control) and 10 (Stress) mice. (**C**) Glucose and insulin tolerance tests before and after 6 weeks of HFD. Values represent the mean ± SEM for 13–14 (Control before HFD), 10–12 (Stress before HFD), 8 (Control after HFD), and 9 (Stress after HFD) mice. * *p* < 0.05 and ** *p* < 0.01 vs. Control before HFD. ^#^
*p* < 0.05 and ^##^
*p* < 0.01 vs. Stress before HFD. ^¶¶^
*p* < 0.01 vs. Control after HFD. (**D**) Serum concentrations of fasting blood glucose and insulin as well as homeostatic model assessment (HOMA)-IR after 6 weeks of HFD. Values represent the mean ± SEM for 8 (Control) and 9 (Stress) mice. * *p* < 0.05 and ** *p* < 0.01 vs. Control. Control, unstressed control mice; Stress, stressed mice; GTT, glucose tolerance test; ITT, insulin tolerance test; AUC, area under the curve.

**Figure 2 cells-09-00996-f002:**
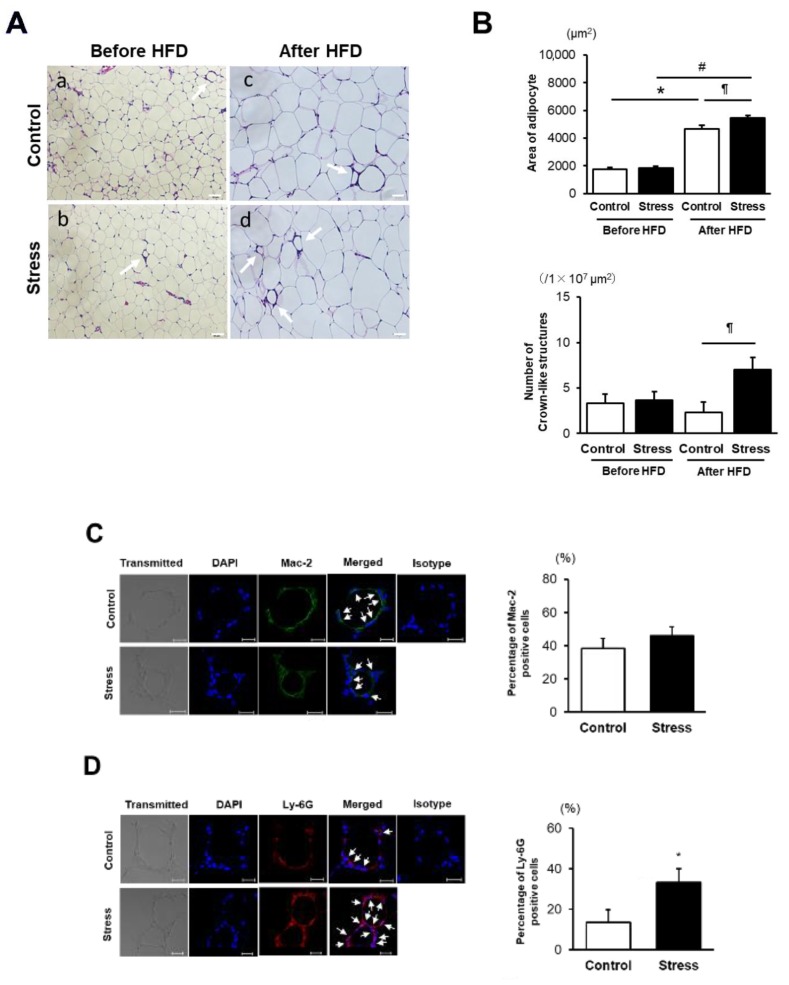
SS enhances neutrophil accumulation in epididymal white adipose tissue (eWAT). (**A**) Representative images of a hematoxylin and eosin-stained eWAT from control (a,c) and stressed (b,d) mice before and after 6 weeks of HFD, respectively. Arrow shows crown-like structures (CLSs). Scale bar = 50 μm. (**B**) Quantitative analysis of adipocyte size and the number of CLSs. Values represent the mean ± SEM for 10 (Control) and 10 (Stress) mice before HFD as well as 10 (Control) and 9 (Stress) mice after 6 weeks of HFD. * *p* < 0.01 vs. Control before HFD. ^#^
*p* < 0.01 vs. Stress before HFD. ^¶^
*p* < 0.05 vs. Control after HFD. (**C**) Immunohistochemical staining of Mac-2-positive cells and the percentages of Mac-2-positive cells in CLSs of eWAT. Values represent the mean ± SEM. Each group consisted of five (Control) and five (Stress) mice. Arrow indicates Mac-2-positive cells. Scale bar = 20 µm. (**D**) Immunohistochemical staining of Ly-6G-positive cells and the percentages of Ly-6G-positive cells in CLSs of eWAT. Values represent the mean ± SEM. Each group consisted of 9 (Control) and 10 (Stress) mice. * *p* < 0.05 vs. Control. Arrow indicates Ly-6G-positive cells. Scale bar = 20 µm. Control, unstressed control mice; Stress, stressed mice.

**Figure 3 cells-09-00996-f003:**
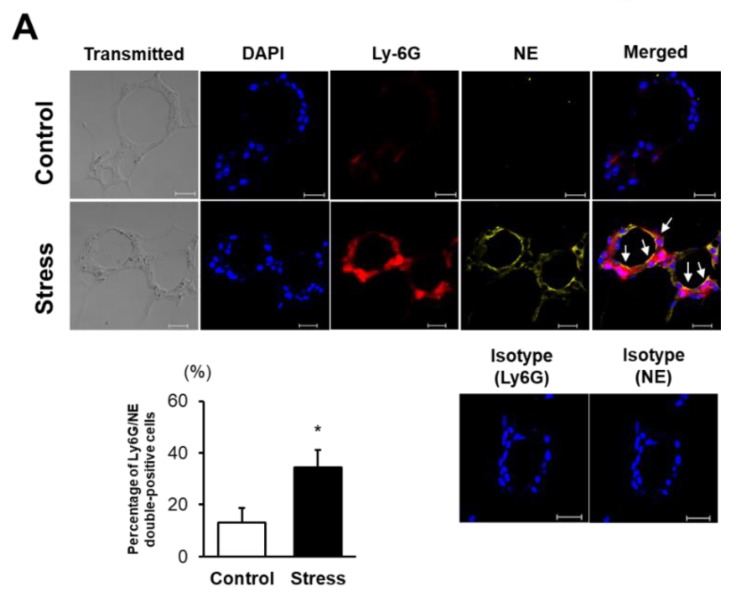
Neutrophil elastase (NE) activation is augmented in eWAT of stressed mice. (**A**) Immunohistochemical staining of Ly-6G/NE-double positive cells and the percentages of Ly-6G/NE-double positive cells in CLSs of eWAT. Values represent the mean ± SEM. Each group consisted of eight (Control) and eight (Stress) mice. * *p* < 0.05 vs. Control. Arrow indicates Ly-6G/NE-double positive cells. Scale bar = 20 µm. (**B**) Quantitative PCR analysis of NE mRNA expression levels in eWAT and liver. Values represent the mean ± SEM relative to Control. Each group consisted of 4 (Control before HFD), 4 (Stress before HFD), 10 (Control after HFD), and 9–10 (Stress after HFD) samples. * *p* < 0.05 vs. Stress before HFD, ^#^
*p* < 0.05 vs. Control after HFD. (**C**) Representative ex vivo images of eWAT and liver as well as quantitative measurement of radiant efficiency corresponding to NE activity. Values represent the mean ± SEM. Each group consisted of 10 (Control) and 10 (Stress) mice. * *p* < 0.05 vs. Control. Control, unstressed control mice; Stress, stressed mice; eWAT, epididymal white adipose tissue; HFD, high-fat diet.

**Figure 4 cells-09-00996-f004:**
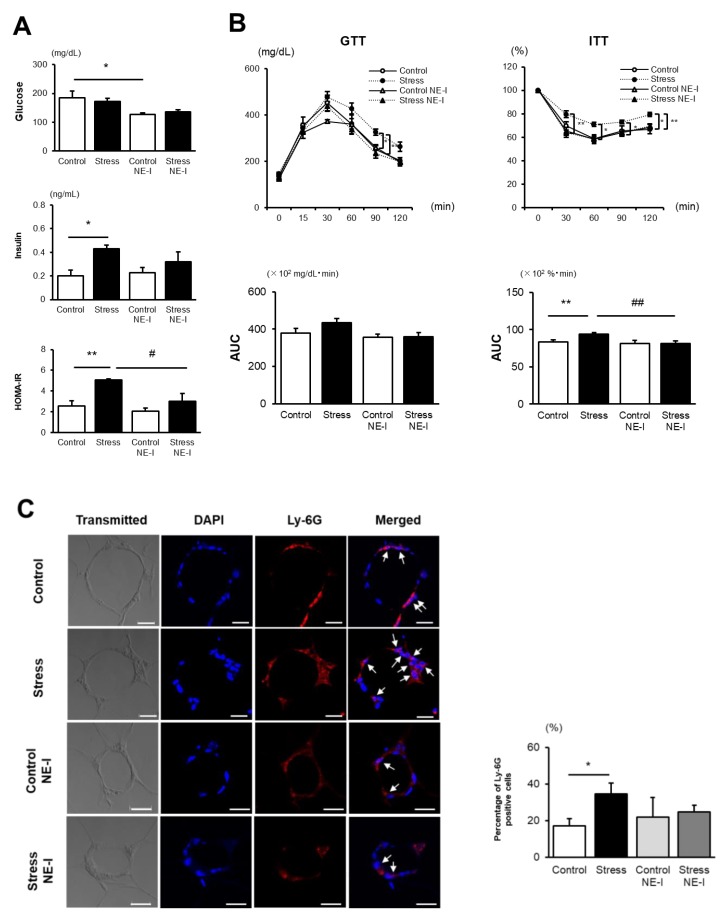
NE inhibitor treatment diminishes the accelerated IR development in stressed mice. (**A**) Serum concentrations of fasting blood glucose and insulin as well as HOMA-IR after 6 weeks of HFD. Values represent the mean ± SEM for four (Control), five (Stress), four (Control NE-I), and four (Stress NE-I) mice. * *p* < 0.05 and ** *p* < 0.01 vs. Control. ^#^
*p* < 0.05 vs. Stress. (**B**) Glucose and insulin tolerance tests after 6 weeks of HFD. Values represent the mean ± SEM for four (Control), five (Stress), four (Control NE-I), and four (Stress NE-I) mice. GTT, glucose tolerance test; ITT, insulin tolerance test, AUC, area under the curve. ** *p* < 0.01 vs. Control. ^##^
*p* < 0.01 vs. Stress. (**C**) Immunohistochemical staining of Ly-6G-positive cells and the percentages of Ly-6G-positive cells in CLSs of eWAT. Values represent the mean ± SEM for four (Control), five (Stress), four (Control NE-I), and four (Stress NE-I) mice. * *p* < 0.05 vs. Control. Arrow indicates Ly-6G-positive cells. Scale bar = 20 µm. Control, unstressed control mice; Stress, stressed mice; Control NE-I, NE inhibitor-treated unstressed control mice; Stress NE-I, NE inhibitor treated stressed mice.

**Figure 5 cells-09-00996-f005:**
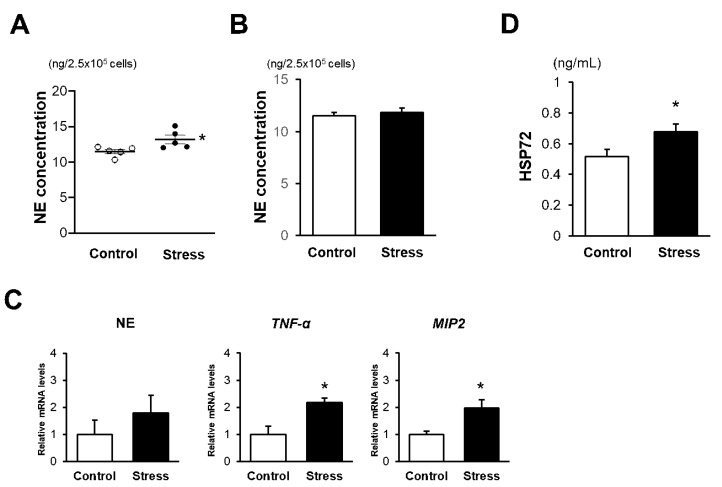
Priming of neutrophils and subsequent NE release upon stimulation are augmented in stressed mice. (**A**) NE concentration in supernatants of BM neutrophils after stimulation with formyl peptide receptor (FPR) 1-specific peptide agonist. Values represent the mean ± SEM. Each group consisted of five (Control) and five (Stress) samples. * *p* < 0.05 vs. Control. (**B**) NE concentration in lysates of bone marrow (BM) neutrophils before stimulation with FPR 1-specific peptide agonist. Values represent the mean ± SEM. Each group consisted of five (Control) and five (Stress) samples. (**C**) Quantitative PCR analysis of NE, TNF-α, and MIP2 mRNA expression levels in peripheral blood neutrophils. Values represent the mean ± SEM relative to Control. Each group consisted of 4–5 (Control) and 4–5 (Stress) samples. * *p* < 0.05 vs. Control. (**D**) Serum concentration of HSP-72 before HFD feeding. Values represent the mean ± SEM. Each group consisted of five (Control) and five (Stress) samples. * *p* < 0.05 vs. Control. Control, unstressed control mice; Stress, stressed mice.

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
