# Peer review of "Social Stress Increases Vulnerability to High-Fat Diet-Induced Insulin Resistance by Enhancing Neutrophil Elastase Activity in Adipose Tissue"

_cells, 2020, doi:10.3390/cells9040996_

Round 1

Reviewer 1 Report

The authors showed that social defeat stress for 10 days increases neutrophil elastase activity in WAT to exacerbate insulin resistance in high-fat-diet-fed mice.

Several questions and deficits are raised to investigate this hypothesis.

As for mice, adipose tissue accounts for ∼15–20% of insulin-stimulated glucose uptake, and skeletal muscle is responsible for ∼80% of whole-body, insulin-mediated glucose metabolism. If neutrophil elastase activity in adipose tissues is involved in PS-induced insulin resistance, how neutrophil elastase affects systemic insulin sensitivity?

The second question is why PS induced neutrophil accumulation in adipose tissue. Is this accumulation is specific to adipose? The authors only checked adipose and liver.

The authors showed that social defeat stress for 10days did not evoke IR in the lean mice. But the author did not testify the various amounts of PS in the lean mice at first.

As the effects of obese and PS are investigated at the same time, it is hard to tell how PS increase neutrophil elastase activity in WAT and induce IR.

Major concerns

1) How neutrophil elastase activity in adipose tissues affects systemic IR?

2) Why PS accumulates neutrophil specifically in adipose tissues? Please check other organs.

3) Please testify the effects of longer periods of PS in the lean mice.

4) The authors should show how PS affects neutrophil elastase activity in adipose tissues in the lean mice, and then compare the results of the lean and fat mice.

Author Response

Response to the Reviewer 1

Thank you very much for your careful reading of our manuscript and helpful comments. In response to your comments, we have performed several additional experiments and revised our manuscript. We hope that this revised manuscript is improved well by your helpful comments.

Reviewer’s Comment # 1

As for mice, adipose tissue accounts for ∼15–20% of insulin-stimulated glucose uptake, and skeletal muscle is responsible for ∼80% of whole-body, insulin-mediated glucose metabolism. If neutrophil elastase activity in adipose tissues is involved in PS-induced insulin resistance, how neutrophil elastase affects systemic insulin sensitivity?

1) How neutrophil elastase activity in adipose tissues affects systemic IR?

Response to the Comment # 1

          As the reviewer mentions, adipose tissue is not a quantitatively significant site for insulin-mediated glucose disposal.

Mechanistic links between NE activation in adipose tissue and systemic insulin resistance remain obscure in this study. Walsh DE et al. reported that activated NE functions as a Toll-like receptor (TLR) 4 activator in human bronchial epithelium (Ref.35 in the text, see below). Likewise, Talukdar et al. also showed that stimulation of intraperitoneal macrophages upon recombinant mouse NE significantly increases TNF-α mRNA expression in the TLR4-dependent manner (Ref.16 in the text, see below). We observed that TNF-α mRNA expression in eWAT was significantly increased in stressed mice after HFD feeding, the extent of which was 3-fold higher than that in HFD-fed control mice (Supplementary Fig. S7). Considering that eWAT macrophages fraction was comparable between the two groups, augmented NE activation in neutrophils is likely to contribute to an increase in TNF-α mRNA expression in eWAT via activation of adipose tissue macrophages (ATMs). Accumulation of ATM and subsequent inflammatory response have emerged as a critical contributor to promote whole-body insulin resistance (Ref.36,37 in the text, see below). TNF-α is one of the most commonly implicated in insulin resistance by inhibiting the effect of insulin to suppress lipolysis in adipocytes (Ref.38,39 in the text, see below). Augmented release of lipolytic substrate such as non-esterified fatty acids (NEFA) is a critical for insulin resistance by impairing insulin-stimulated glucose uptake (Ref.40,41 in the text, see below). However, the relative contributions of NEFA to skeletal muscle IR are less elucidated than hepatic IR (Ref.42 in the text, see below), and therefore, integrated mechanisms linking between lipolysis in adipose tissue and stress-associated whole-body IR need to be investigated in future studies.  

Reference

  1. Walsh, D.E.; Greene, C.M.; Carroll, T.P.; Taggart, C.C.; Gallagher, P.M.; O'Neill, S.J.; McElvaney, N.G. Interleukin-8 up-regulation by neutrophil elastase is mediated by MyD88/IRAK/TRAF-6 in human bronchial epithelium. J Biol Chem 2001, 276, 35494-35499.
  2. Talukdar, S.; Oh, D.Y.; Bandyopadhyay, G.; Li, D.; Xu, J.; McNelis, J.; Lu, M.; Li, P.; Yan, Q.; Zhu, Y.; Ofrecio, J.; Lin, M.; Brenner, M.B.; Olefsky, J.M. Neutrophils mediate insulin resistance in mice fed a high-fat diet through secreted elastase. Nat Med 2012, 18, 1407-1412, doi:10.1038/nm.2885.
  3. Olefsky, J.M.; Glass, C.K. Macrophages, inflammation, and insulin resistance. Annu Rev Physiol 2010, 72, 219-46, doi: 10.1146/annurev-physiol-021909-135846.
  4. Wernstedt, A.I.; Tao, C.; Morley, T.S.; Wang, Q.A.; Delgado-Lopez, F.; Wang, Z.V.; Scherer, P.E. Adipocyte inflammation is essential for healthy adipose tissue expansion and remodeling. Cell Metab 2014, 20, 103-118, doi: 10.1016/j.cmet.2014.05.005.
  5. Hotamisligil, G.S.; Shargill, N.S.; Spiegelman, B.M. Adipose expression of tumor necrosis factor-alpha: direct role in obesity-linked insulin resistance. Science 1993, 259, 87-91.
  6. Uysal, K.T.; Wiesbrock, S.M.; Marino, M.W.; Hotamisligil, G.S. Protection from obesity-induced insulin resistance in mice lacking TNF-alpha function. Nature 1997, 389, 610-614.
  7. Laurencikiene, J.; van Harmelen, V.; Arvidsson, N.E.; Dicker, A.; Blomqvist, L.; Näslund, E.; Langin, D.; Arner, P.; Rydén, M. NF-kappaB is important for TNF-alpha-induced lipolysis in human adipocytes. J Lipid Res 2007,48, 1069-1077.
  8. Ranjit, S.; Boutet, E.; Gandhi, P.; Prot, M.; Tamori, Y.; Chawla, A.; Greenberg, A.S.; Puri, V.; Czech, M.P. Regulation of fat specific protein 27 by isoproterenol and TNF-α to control lipolysis in murine adipocytes. J Lipid Res 2011;52:221-236. doi: 10.1194/jlr.M008771.
  9. Petersen, M.C.; Shulman, G.I. Mechanisms of Insulin Action and Insulin Resistance. Physiol Rev 2018, 98, 2133-2223, doi: 10.1152/physrev.00063.2017.

These were stated in line 452 page 14 in the Revised Discussion section.

Reviewer’s Comment # 2

The second question is why PS induced neutrophil accumulation in adipose tissue. Is this accumulation is specific to adipose? The authors only checked adipose and liver.

2) Why PS accumulates neutrophil specifically in adipose tissues? Please check other organs.

Response to the Comment # 2

To respond to the Reviewer’s important comment, we also examined the effect of SS [PS was rephrased to social stress (SS) according to the comment of Reviewer 3] on the NE activity in the lower limbs in which skeletal muscle was involved. However, there was no difference in NE activity between the two groups (Supplementary Fig. S7).

Elgazar-Carmon V et al. reported that neutrophils transiently infiltrate the parenchyma of intra-abdominal adipose tissue of HFD-fed mice and also observed that a physical binding between neutrophils and adipocytes using immunohistochemistry analysis (Ref.33 in the text, see below). Consistent with in vivo findings, they demonstrated that adherence of mouse peritoneal neutrophils to a monolayer of 3T3-L1 mouse adipocytes was significantly exaggerated after stimulation of neutrophils upon IL-8 analogue CXCL1 chemokine. They also showed that the adherence of neutrophils was prevented by preincubating with anti-ICAM-1 antibodies. Furthermore, Brake DK et al. reported that ICAM-1 mRNA expression was significantly increased in the abdominal fat of HFD-fed mice in a tissue-specific manner (Ref.34 in the text, see below). Considering that ICAM-1 expression in eWAT was comparable between stressed and control mice (Supplementary Fig. S7), and that CXCL2/MIP2 (murine IL-8 homologue) expression was significantly elevated in peripheral blood neutrophils of stressed mice, SS-induced priming of neutrophils is likely to contribute to the eWAT-specific enhancement of NE activation in stressed mice.

Reference

  1. Elgazar-Carmon, V.; Rudich, A.; Hadad, N.; Levy, R. Neutrophils transiently infiltrate intra-abdominal fat early in the course of high-fat feeding. J Lipid Res 2008, 49, 1894-1903, doi: 10.1194/jlr.M800132-JLR200.
  2. Brake, D.K.; Smith, E.O.; Mersmann, H.; Smith, C.W.; Robker, R.L. ICAM-1 expression in adipose tissue: effects of diet-induced obesity in mice. Am J Physiol Cell Physiol 2006, 291, C1232-1239.

These were stated in line 298 page 8 in the Revised Results section and in line  439 page 13 in the Revised Discussion section.

Reviewer’s Comment # 3

The authors showed that social defeat stress for 10 days did not evoke IR in the lean mice. But the author did not testify the various amounts of PS in the lean mice at first.

3) Please testify the effects of longer periods of PS in the lean mice.

Response to the Comment # 3

          Although we did not examine the effects of longer periods of SS on insulin resistance in the lean mice, we examined the effect of repeated social defeat (RSD) that allowed for direct physical contact for 10 min each day during the 10 days of SS, leading to the development of depression-like behavior (Ref. 18 in the text, see below). Social interaction ratio of RSD-exposed mice was significantly lower than those of control and SS-exposed mice (0.77 ± 0.11, p < 0.05); however, there was no difference in glucose and insulin tolerance between the three groups (Supplementary Fig. S4), suggesting that various amount of SS in the lean mice is not likely to evoke IR.

Reference

  1. Golden, S.A.; Covington, H.E. 3rd.; Berton, O.; Russo, S.J. A standardized protocol for repeated social defeat stress in mice. Nat Protoc 2011, 6, 1183-1191, doi:1038/nprot.2011.361.

These were stated in line 236 page 6 in the Revised Results section.

Reviewer’s Comment # 4

As the effects of obese and PS are investigated at the same time, it is hard to tell how PS increase neutrophil elastase activity in WAT and induce IR.

4) The authors should show how PS affects neutrophil elastase activity in adipose tissues in the lean mice, and then compare the results of the lean and fat mice.

Response to the Comment # 4

          Unfortunately, we had not examined ex vivo NE activity before HFD feeding. To respond to the Reviewer’s important comment, we performed RT-PCR analysis of eWAT and liver before HFD feeding as shown in Fig. 3B. NE mRNA expression levels in eWAT were comparable between the two groups before HFD feeding; however, they were significantly increased in eWAT of stressed mice after HFD feeding, the extent of which was much higher than those in HFD-fed control mice, whereas NE mRNA expression levels in liver were comparable before and after HFD feeding in both groups of mice, suggesting the possibility that neutrophil accumulation before HFD feeding was not affected by SS and that enhanced neutrophil accumulation and subsequent NE activation was exhibited after HFD feeding.         

These were stated in line 290 page 8 in the Revised Results section.

All authors are very grateful to Reviewer 1 for carefully reviewed comments and suggestions. We strongly hope that this revised version will be acceptable by the Reviewer 1.

Reviewer 2 Report

A brief summary:

This is an interesting paper investigating if changes in neutrophils after psychological stress can increase the development of high fat diet induced insulin resistance in male mice. Furthermore, the authors perform an intervention by treating the mice with a neutrophil elastase inhibitor and thereby succeed in abrogating the insulin sensitivity impairment of the stressed mice. The novelty is that they show the link between psychological stress, increased neutrophil elastase and high fat diet induced insulin resistance and that it is possible to block the stress induced insulin resistance by treatment with neutrophil elastase inhibitor.

Broad comments:

  1. The different methods are generally well described. However, an overview (in text or as a schematic figure) of the design of the different experiments and at what age the mice were euthanized for the different analysis would increase the readability of the paper.

  1. The discussion is relevant, but it would be valuable to add some comparison between the mice receiving neutrophil elastase (NE) inhibitor and other NE deficient models such as NE KO mice or alpha-1 antitrypsin transgenic mice, regarding stress response and diet induced insulin resistance.

Specific comments:

  1. Line 25: Exchange wild-type mice to C57BL/6 mice, to clarify that the experimental mice are from another strain than the CD-1 intruder mouse.
  2. Line 80-82: Describe which substance the control mice received during the oral dosing in the NE inhibitor experiment.
  3. Line 84: Clarify if the mice were fasted before they were euthanized and at what time of the day they were euthanized, since the feeding status and circadian rhythm can affect the protein levels of for example insulin and corticosterone in the serum.
  4. Line 108: Were the mice fasted before blood collection for the noradrenalin, corticosterone, insulin and HSP72 analysis, and at what age or stage in the experiment were these samples collected?
  5. Line 131: Clarify at what age or stage in the experiment these samples were collected.
  6. Line 134: The extraction of liver mRNA should also be mentioned here.
  7. Line 201: I suppose it should be “A p-value below 0.05 was considered statistically significant.“ (instead of above 0.05)?
  8. Figure 1D: The figure is correct but it’s confusing that the upper diagram shows the stressed group with white markers while the lower AUC bar graphs shows the stressed group by black bars. It’s preferable to keep the groups marked by the same color at least within the same figure.
  9. Figure S2: According to the figure legend, cumulative food intake was done in 9 mice per group, but they were group housed 3 mice per cage, so the number of statistical observations should be 3 (one measurement per cage not per mouse). Were the statistics calculated for n=3 or n=9?

Author Response

Response to the Reviewer 2

Thank you very much for your careful reading of our manuscript and helpful comments. In response to your comments, we have performed several additional experiments and revised our manuscript. We hope that this revised manuscript is improved well by your helpful comments.

Broad comments:

Reviewer’s Comment # 1

The different methods are generally well described. However, an overview (in text or as a schematic figure) of the design of the different experiments and at what age the mice were euthanized for the different analysis would increase the readability of the paper.

Response to the Comment # 1

          According to the Reviewer’s important comment, the scheme of the design of the different experiments was shown in Supplementary Fig. S2.

These were stated in line 89 2nd page in the Revised Materials and Methods section.

Reviewer’s Comment # 2

The discussion is relevant, but it would be valuable to add some comparison between the mice receiving neutrophil elastase (NE) inhibitor and other NE deficient models such as NE KO mice or alpha-1 antitrypsin transgenic mice, regarding stress response and diet induced insulin resistance.

Response to the Comment # 2

          According to the Reviewer’s important comment, we described the association between stress response and diet-induced insulin resistance in NE deficient mice and alpha-1 antitrypsin transgenic mice as below.

As well as the pharmacological inhibition of NE, genetic deletion of NE in mice exerted more insulin sensitive than wild-type mice (Ref.16 in the text, see below). Consistently, α-1 antitrypsin transgenic mice were resistant to HFD-induced IR (Ref.17 in the text, see below); however, few studies have focused on the association between stress response and HFD-induced IR. To elucidate the causal effect of stress-mediated neutrophil NE activation on HFD-induced IR, future studies using neutrophil-specific conditional knockout mice are needed.     

Reference

  1. Talukdar, S.; Oh, D.Y.; Bandyopadhyay, G.; Li, D.; Xu, J.; McNelis, J.; Lu, M.; Li, P.; Yan, Q.; Zhu, Y.; Ofrecio, J.; Lin, M.; Brenner, M.B.; Olefsky, J.M. Neutrophils mediate insulin resistance in mice fed a high-fat diet through secreted elastase. Nat Med 2012, 18, 1407-1412, doi:10.1038/nm.2885.
  2. Mansuy-Aubert, V.; Zhou, Q.L.; Xie, X.; Gong, Z.; Huang, J.Y.; Khan, A.R.; Aubert, G.; Candelaria, K.; Thomas, S.; Shin, D.J.; Booth, S.; Baig, S.M.; Bilal, A.; Hwang, D.; Zhang, H.; Lovell-Badge, R.; Smith, S.R.; Awan, F.R.; Jiang, Z.Y. Imbalance between neutrophil elastase and its inhibitor α1-antitrypsin in obesity alters insulin sensitivity, inflammation, and energy expenditure. Cell Metab 2013, 17, 534-548, doi:10.1016/j.cmet.2013.03.005.

These were stated in line 477 page 14 in the Revised Discussion section.

Specific comments:

Reviewer’s Comment # 3

Line 25: Exchange wild-type mice to C57BL/6 mice, to clarify that the experimental mice are from another strain than the CD-1 intruder mouse.

Response to the Comment # 3

          To respond to the Reviewer’s important comment, we exchanged wild-type mice to C57BL/6 mice in line 25 in the Abstract section.

Thank you for your careful attention.

Reviewer’s Comment # 4

Line 80-82: Describe which substance the control mice received during the oral dosing in the NE inhibitor experiment.

Response to the Comment # 4

          According to the Reviewer’s important comment, we described that control mice were given vehicle (distilled water) in the same manner.

These were stated in line 85 2nd page in the Revised Materials and Methods section.

Reviewer’s Comment # 5

Line 84: Clarify if the mice were fasted before they were euthanized and at what time of the day they were euthanized, since the feeding status and circadian rhythm can affect the protein levels of for example insulin and corticosterone in the serum.

Response to the Comment # 5

          According to the Reviewer’s important comment, we described that blood samples were obtained in the morning (8 am - 10 am) after an overnight fast.

These were stated in line 88 2nd page in the Revised Materials and Methods section.

Reviewer’s Comment # 6

Line 108: Were the mice fasted before blood collection for the noradrenalin, corticosterone, insulin and HSP72 analysis, and at what age or stage in the experiment were these samples collected?

Response to the Comment # 6

          According to the Reviewer’s important comment, we described that blood samples were obtained in the morning (8 am - 10 am) after an overnight fast. Serum levels of noradrenaline, corticosterone, and HSP72 were examined in 10-week-old mice before HFD feeding and serum insulin level was examined in 16-week-old mice after 6 weeks of HFD feeding.

These were stated in line 115 page 3 in the Revised Materials and Methods section.

Reviewer’s Comment # 7

Line 131: Clarify at what age or stage in the experiment these samples were collected.

Response to the Comment # 7

          According to the Reviewer’s important comment, we described that blood samples were obtained in the morning (8 am - 10 am) after an overnight fast in 10-week-old mice before HFD feeding and in 16-week-old mice after 6 weeks of HFD feeding.

These were stated in line 141 page 4 in the Revised Materials and Methods section.

Reviewer’s Comment # 8

Line 134: The extraction of liver mRNA should also be mentioned here.

Response to the Comment # 8

According to the Reviewer’s important comment, we described the extraction of liver mRNA.

These were stated in line 146 page 4 in the Revised Materials and Methods section.

Thank you for your careful reading.

Reviewer’s Comment # 9

Line 201: I suppose it should be “A p-value below 0.05 was considered statistically significant.“ (instead of above 0.05)?

Response to the Comment # 9

We corrected this typographic error in line 214 page 5 in the Revised Materials and Methods section. We would like to thank you very much for your careful attention.

Reviewer’s Comment # 10

Figure 1D: The figure is correct but it’s confusing that the upper diagram shows the stressed group with white markers while the lower AUC bar graphs shows the stressed group by black bars. It’s preferable to keep the groups marked by the same color at least within the same figure.

Response to the Comment # 10

According to the Reviewer’s important comment, we revised the Figure 1D (Figure 1C in the revised text) to make it easy to see. Thank you very much for your careful attention.

Reviewer’s Comment # 11

Figure S2: According to the figure legend, cumulative food intake was done in 9 mice per group, but they were group housed 3 mice per cage, so the number of statistical observations should be 3 (one measurement per cage not per mouse).

Were the statistics calculated for n=3 or n=9?

Response to the Comment # 11

According to the Reviewer’s important comment, we revised the Supplementary Figure Legends in Fig. S2 (Figure S3 in the revised text) described below.

(C) Cumulative caloric intake over 6 weeks was equivalent between the two groups. Mice were group housed 3 mice per cage. Values represent the mean ± SEM for 3 (Control) and 3 (Stress) cages. Control, unstressed control mice; Stress, stressed mice.

All authors are very grateful to Reviewer 2 for carefully reviewed comments and suggestions. We strongly hope that this revised version will be acceptable by the Reviewer 2.

Reviewer 3 Report

Manuscript ID: cells-726225

Title: Psychological stress increases vulnerability to high-fat diet-induced  insulin resistance by enhancing neutrophil elastase activity in adipose tissue

Authors: Shinichiro Motoyama, et al

Summary:

This is an interesting study showing that “Psychological Stress” (rather “Social Stress” see below) may have an impact on HFD-induced IR through the regulation of neutrophil-produced elastase activity. The main concerns of this reviewer  are based on the degree of development of IR after the HFD, the statement that IR is “accelerated” in stressed animals when no dynamics over time are shown and the adscription of effects to locally produced elastase in eWAT when elastase inhibitor treatment was orally delivered and, thus, systemically distributed. Otherwise, results are overall clear and interesting, the experimental design and techniques are adequate and conclusions are overall sound.

MAJOR POINTS

  • One of the main concerns with this study is that the final degree of IR achieved after the HFD feeding is unclear to me, what is a key point for the conclusions of this study, at least from what can be seen comparing the graphs in Fig 1D and that in Suppl F1. In fact, the ITTs shown for the standard diet fed mice also appear to have a not very healthy insulin sensitivity, maybe because of the animal facility conditions, inbreeding or other causes compared to published data from the same strain. The authors should show the results from the two groups of animals (stressed and controls) both before and after the HFD in the same graph and discuss the results (and, if possible, differences with published data). The AUC of standard diet-fed mice should be compared to that of HFD-fed animals and statistical analysis performed to clearly state IR has been established.
  • Related to this, statements such as “… exposure to PS accelerates the development of HFD-induced IR,..:” (pp5, line 215) ; “Figure 1. Psychological stress (PS) accelerates the development of high-fat diet (HFD)-induced 219 insulin resistance (IR).” (pp6, line 219); “augmented accumulation of 233 inflammatory cells may contribute to the early onset of IR development in stressed mice…” (pp 6, line 234) could only be used if the control group of mice is demonstrated to finally develop IR but later on in this process (after 8 weeks of HFD?). However, no data are shown to be able to state this fact rigorously, and this supposedly “accelerated” process cannot be compared to the “regular speed” one. So, the acceleration remains to be proven. These data should be included or, alternatively, sentences regarding these issues should be rephrased throughout the MS, particularly in the Abstract (“…PS accelerates the development of HFD-induced IR”).
  • In the same line, do the authors know what happens to IR differences between control and PS mice after an 8 week HFD? Are differences maintained or are they present only at earlier (6 weeks) time points like in Liu L et al. [10] in chronic noise exposure models?
  • In section 6. Immunohistochemistry it is stated thatThe percentage of Mac2-stained 127 nuclei in total nuclei number of crown-like structures was assessed per one section from 5 animals 128 from each group.The same holds true for Ly-6G- and Ly-6G/NE staining in Figs 3A and 4C that is essential for the paper. Does this mean that only one section was quantified per animal? If so, this number should be increased to include more sections per mice and statistics should be redone. Also, please specify how precisely representative sections were chosen for analysis (random?) and how many. These data are very relevant for the conclusions drawn and a sound statistical analysis is important here.
  • Given that the actual difference, even when significant, of noradrenaline and (of particular importance) of elastase between control and stressed groups is very narrow, please provide graphs for panels 1A and 5A that show data not as bar graphs but rather include all individual points from each mice (as well as mean and statistics) to be able to visually assess for dispersion of results.
  • Also, statements related to treatment with orally administered elastase inhibitors like “These findings strongly support the notion that enhanced NE activation in eWAT profoundly contributes to augmented IR development in stressed mice.” (pp9, line 283) should be rephrased. Even when the authors show lower percentage of Ly-6G-positive cells in CLS from eWAT after NE inhibitor treatment, the effect of elastase inhibition in other tissues (where it will be biodistributed after oral administration) cannot be completely ruled out and care should be applied to state this in an accurate manner. Please, rephrase throughout the manuscript.
  • Maybe it would be more accurate to substitute the term “Psychological Stress” for “Social Stress” or “Social Disruption” in the MS and title. First to differentiate the source and intensity of the stress induced. Secondly, since most of the references in the field (including those cited by the authors) use that definition for the experimental setting used here. In other case, please explain why PS would be more appropriate here.

MINOR POINTS

  • One of the main issues dealing with this study deals with the net influence of stress relative to other variables in the differences evaluated. In particular, the resident mouse was separated from the intruder animal by a perforated partition, which “allowed for continuous visual, auditory, and olfactory contact with no physical interaction”. Given that they shared cage and, I assume, food pellets and bed/shaving inside the cage, how can this experimental setting guarantee that microbiota was not passed on from the CD-1 mouse to the C57? As described, microbiota profile has a very important and clear impact on IR development in particular after a HFD. How sure are you that the bed was not shared between compartments inside the same cage? What about feces (mice often eat them during fasting periods)? Do the parameters of C57 mice that have been caged using CD-1 beds (but without having been actually exposed to CD-1 animals) also change?
  • The eWAT weight/BW seems to be indistinguishable between control and stressed groups even when the levels of noradrenaline in the stressed animals are significantly higher. An explanation should be provided for this apparent discrepancy and the weight of adipose depots (not normalized to BW) should be shown.
  • Care should be taken when expressing average food intake per cage in cages with 3 similarly aged mice. The individual variations may be masked by the mean intake (taken as individual value). And, in particular, in experimental settings dealing with stress, the stress level may vary greatly in one individual versus another. The ideal scenario would have been that food intake was measured for each mice. However, I understand this would require individual caging and subsequent impact on stress. So, I only ask that his point is at least clarified in the text.
  • Are the resident mice always caged in groups of 3? This is unclear from the text that seems to say C57 animals are exposed one-to-one to CD-1 mice “… After screening of aggressor CD-1 mice, each CD-1 resident mouse received a wild-type intruder mouse and the two animals were separated by a perforated partition”
  • Related to this, to make data more representative, photographs in Panel 2A should include lower magnification fields showing the presence of not only one but if possible several crown-like structures, even when adipocyte size is decreased in the final picture. Same for pictures 3A and 4C.
  • In section “2.12. Statistical analysis:”… followed by a Tukey-Kramer test to analyze significant differences between the groups. A p-value above05 was considered statistically significant.” Do you mean BELOW 0.05? Is this test different form other p-value based analysis or is this a typo?
  • Please specify how precisely cumulative caloric intake was measured for Suppl Fig 2C
  • It would be advisable to discuss the conclusions drawn from differences observed between the increased ex vivo NE activity in eWAT while no differences were found in liver. Since the latter organ is key for the maintenance of glucose homeostasis and IR.
  • Please check English throughout the MS to correct and avoid frequent errors such as:

-“… Blood glucose concentrations were measured at 0 min before and 15, 30, 60, 90, and 120 min after…”

-“… HSP72 were estimated using an ELISA kits (BA E-5200; Labor Diagnostika Nord, Nordhorn,  Germany; ab108821; Abcam plc, Cambridge, UK;…)”.

-“…2.11. In vitro activation of bone marrow (BM) neutrophils on formyl peptide receptor (FPR) 1”

-“…The 680 FAST comprises of two near-infrared 176 (NIR) fluorochromes…”

-“…noradrenalin­­_“ (pp6 Figure 1 legend)

-Clarify what is meant by “these 344 findings are not likely the case with the pathogenesis of IR in the clinical settings of metabolic 345 syndrome [5,6]. “ (pp12, line 345).

among others,

Author Response

Response to the Reviewer 3

Thank you very much for your careful reading of our manuscript and helpful comments. In response to your comments, we have performed several additional experiments and revised our manuscript. We hope that this revised manuscript is improved well by your helpful comments.

Reviewer’s Comment # 1

One of the main concerns with this study is that the final degree of IR achieved after the HFD feeding is unclear to me, what is a key point for the conclusions of this study, at least from what can be seen comparing the graphs in Fig 1D and that in Suppl F1. In fact, the ITTs shown for the standard diet fed mice also appear to have a not very healthy insulin sensitivity, maybe because of the animal facility conditions, inbreeding or other causes compared to published data from the same strain. The authors should show the results from the two groups of animals (stressed and controls) both before and after the HFD in the same graph and discuss the results (and, if possible, differences with published data). The AUC of standard diet-fed mice should be compared to that of HFD-fed animals and statistical analysis performed to clearly state IR has been established.

Response to the Comment # 1

          We completely agree with the Reviewer’s important comment. To correctly respond to the Reviewer’s comment, we showed the results from the two groups of animals (stressed and controls) both before and after the HFD in the same graph as shown in Fig. 1C (Fig. 1D in the original text). Glucose tolerance was significantly impaired in HFD-fed mice compared with those in mice before HFD feeding; however, there was no difference between the two groups of control and stressed mice. In contrast, insulin sensitivity in stressed mice was significantly impaired after HFD feeding, resulting in the significantly higher AUC than HFD-fed control mice (Fig. 1C). Consistently, serum insulin levels and homeostasis model assessment (HOMA)-IR after HFD feeding were significantly higher in stressed mice than in control mice (Fig. 1D). These findings indicate that IR development was established in stressed mice after 6 weeks of HFD feeding.

We also added the sentence about statistical analysis as below. Significant differences among groups for dependent variables were detected using two-way ANOVA: SS [PS was rephrased to social stress (SS) according to the Reviewer’s comment] (Control versus Stress), diet (before HFD feeding versus after HFD feeding).

These were stated in line 212 page 5 in the Revised Materials and Methods section, in line 222 page 5 in the Revised Results section, and as shown in Figure 1C.

Reviewer’s Comment # 2

Related to this, statements such as “… exposure to PS accelerates the development of HFD-induced IR,..:” (pp5, line 215) ; “Figure 1. Psychological stress (PS) accelerates the development of high-fat diet (HFD)-induced 219 insulin resistance (IR).” (pp6, line 219); “augmented accumulation of 233 inflammatory cells may contribute to the early onset of IR development in stressed mice…” (pp 6, line 234) could only be used if the control group of mice is demonstrated to finally develop IR but later on in this process (after 8 weeks of HFD?). However, no data are shown to be able to state this fact rigorously, and this supposedly “accelerated” process cannot be compared to the “regular speed” one. So, the acceleration remains to be proven. These data should be included or, alternatively, sentences regarding these issues should be rephrased throughout the MS, particularly in the Abstract (“…PS accelerates the development of HFD-induced IR”).

Response to the Comment # 2

          We completely agree with the Reviewer’s important comment. We rephrased these sentences in the revised manuscript as below.

1) Our findings show that SS-exposed mice are susceptible to the development of HFD-induced IR accompanied by augmented NE activity. (line 35 1st page in Abstract)

2) 3.1. SS increases the vulnerability to the development of HFD-induced IR (line 217 page 5)

3) These findings show that SS-exposed mice are susceptible to the development of HFD-induced IR, which is independent of food intake and BW gain. (line 243 page 6)

4) Figure 1. Social stress (SS) increases the susceptibility to the development of high-fat diet (HFD)-induced insulin resistance (IR). (line 247 page 6 in Figure legend)

5) These findings support the notion that enhanced NE activation in eWAT is closely related to the augmented IR development in stressed mice. (line 321 page 10)

6) The results of our study show that SS activates the SNS and subsequently increases the vulnerability to HFD-induced IR, accompanied by a marked increase in NE activity in eWAT. (line 470 page 14)

Reviewer’s Comment # 3

In the same line, do the authors know what happens to IR differences between control and PS mice after an 8 week HFD? Are differences maintained or are they present only at earlier (6 weeks) time points like in Liu L et al. [10] in chronic noise exposure models?

Response to the Comment # 3

          Unfortunately, we did not examine the effect of SS on the late phase of IR development. The first infiltrating inflammatory cells after starting of HFD are neutrophils, but not macrophages, we therefore focused on the early onset of HFD-induced IR to examine the contribution of SS-induced activation of neutrophils. Talukdar S et al. showed that augmented accumulation of neutrophils in a few days after HFD feeding was still observed up to 90 days of HFD feeding (Ref. 16 in the text, see below). Because the neutrophils are involved in the recruiting and activating ATMs during high-fat feeding, the effect of SS-induced activation of neutrophils on the late phase of IR development need to be investigated in future studies.

  1. Talukdar, S.; Oh, D.Y.; Bandyopadhyay, G.; Li, D.; Xu, J.; McNelis, J.; Lu, M.; Li, P.; Yan, Q.; Zhu, Y.; Ofrecio, J.; Lin, M.; Brenner, M.B.; Olefsky, J.M. Neutrophils mediate insulin resistance in mice fed a high-fat diet through secreted elastase. Nat Med 2012, 18, 1407-1412, doi:10.1038/nm.2885.

These were stated in line 395 page 13 in the Revised Discussion section.

Reviewer’s Comment # 4

In section 6. Immunohistochemistry it is stated that “The percentage of Mac2-stained nuclei in total nuclei number of crown-like structures was assessed per one section from 5 animals from each group.” The same holds true for Ly-6G- and Ly-6G/NE staining in Figs 3A and 4C that is essential for the paper. Does this mean that only one section was quantified per animal? If so, this number should be increased to include more sections per mice and statistics should be redone. Also, please specify how precisely representative sections were chosen for analysis (random?) and how many. These data are very relevant for the conclusions drawn and a sound statistical analysis is important here.

Response to the Comment # 4

          Immunohistological staining was performed at least 3 sections from each animal and the total number of 4 to 6 representative images were chosen from each animals at random, and then analyzed. We corrected the sentence as below.

The percentages of Ly-6G- and Ly-6G/NE-positive stained nuclei in total number of crown-like structures was assessed per 3 sections from 8-10 animals from each group. The percentage of Mac2-stained nuclei in total nuclei number of crown-like structures was assessed per 3 sections from 5 animals from each group. The total number of 4 to 6 representative images were chosen from each animals at random, and then analyzed.

These were stated in line 134 page 3 in the Revised Materials and Methods section.

Reviewer’s Comment # 5

Given that the actual difference, even when significant, of noradrenaline and (of particular importance) of elastase between control and stressed groups is very narrow, please provide graphs for panels 1A and 5A that show data not as bar graphs but rather include all individual points from each mice (as well as mean and statistics) to be able to visually assess for dispersion of results.

Response to the Comment # 5

          To respond to the Reviewer’s important comment, we revised Fig. 1A and Fig. 5A showing all individual data.

Reviewer’s Comment # 6

Also, statements related to treatment with orally administered elastase inhibitors like “These findings strongly support the notion that enhanced NE activation in eWAT profoundly contributes to augmented IR development in stressed mice.” (pp9, line 283) should be rephrased. Even when the authors show lower percentage of Ly-6G-positive cells in CLS from eWAT after NE inhibitor treatment, the effect of elastase inhibition in other tissues (where it will be biodistributed after oral administration) cannot be completely ruled out and care should be applied to state this in an accurate manner. Please, rephrase throughout the manuscript.

Response to the Comment # 6

          We completely agree with the Reviewer’s precise comment. We rephrased throughout the manuscript as below.

1) These findings support the notion that enhanced NE activation in eWAT is closely related to the augmented IR development in stressed mice. (line 321 page 10)

2) To elucidate the causal effect of stress-mediated neutrophil NE activation on HFD-induced IR, future studies using neutrophil-specific conditional knockout mice are needed. (line 480 page 14)

Reviewer’s Comment # 7

Maybe it would be more accurate to substitute the term “Psychological Stress” for “Social Stress” or “Social Disruption” in the MS and title. First to differentiate the source and intensity of the stress induced. Secondly, since most of the references in the field (including those cited by the authors) use that definition for the experimental setting used here. In other case, please explain why PS would be more appropriate here.

Response to the Comment # 7

          According to the Reviewer’s important comment, we rephrased “Psychological Stress” to “Social Stress”.

MINOR POINTS

Reviewer’s Comment # 8

One of the main issues dealing with this study deals with the net influence of stress relative to other variables in the differences evaluated. In particular, the resident mouse was separated from the intruder animal by a perforated partition, which “allowed for continuous visual, auditory, and olfactory contact with no physical interaction”. Given that they shared cage and, I assume, food pellets and bed/shaving inside the cage, how can this experimental setting guarantee that microbiota was not passed on from the CD-1 mouse to the C57? As described, microbiota profile has a very important and clear impact on IR development in particular after a HFD. How sure are you that the bed was not shared between compartments inside the same cage? What about feces (mice often eat them during fasting periods)? Do the parameters of C57 mice that have been caged using CD-1 beds (but without having been actually exposed to CD-1 animals) also change?

Response to the Comment # 8

          Thank you very much for your meaningful comments. As the reviewer mentions, microbiota is extremely critical for the development of diet-induced insulin resistance. To respond to the Reviewer’s important comment, the pictures and detail description of perforated partition were added. The partition was solid and tightly fixed in the bottom of the cage and the size of perforation was small and located 3 cm above the bottom. Further, during the 10 days of SS, this perforated partition was not removed. Thus, it is unlikely to share the food pellets and feces between compartments inside the same cage (Supplementary Fig. S1).

These were stated in line 75 2ndpage in the Revised Materials and Methods section.

Reviewer’s Comment # 9

The eWAT weight/BW seems to be indistinguishable between control and stressed groups even when the levels of noradrenaline in the stressed animals are significantly higher. An explanation should be provided for this apparent discrepancy and the weight of adipose depots (not normalized to BW) should be shown.

Response to the Comment # 9

          To respond to the Reviewer’s important comment, we added the data of the weight of adipose depots (not normalized to BW) in Supplementary Fig. 3B (Supplementary Fig. 2B in the original text).

The eWAT weight and eWAT weight/BW in stressed mice tended to be lower than those in control mice, but not significant (p < 0.09, p < 0.07 vs. control, respectively). Actual difference in serum noradrenaline level was very narrow, which was considered as a possible explanation why eWAT weight and eWAT weight/BW did not reach statistical difference.

These were stated in line 232 page 5 in the Revised Results section.

Reviewer’s Comment # 10

Care should be taken when expressing average food intake per cage in cages with 3 similarly aged mice. The individual variations may be masked by the mean intake (taken as individual value). And, in particular, in experimental settings dealing with stress, the stress level may vary greatly in one individual versus another. The ideal scenario would have been that food intake was measured for each mice. However, I understand this would require individual caging and subsequent impact on stress. So, I only ask that this point is at least clarified in the text.

Response to the Comment # 10

          According to the Reviewer’s important comment, we described in the Results section as below.

However, taking into consideration that average food intake per cage may mask the individual variations responsible for their stress levels, this finding need to be interpreted with caution.

These were stated in line 230 page 5 in the Revised Results section.

Reviewer’s Comment # 11

Are the resident mice always caged in groups of 3? This is unclear from the text that seems to say C57 animals are exposed one-to-one to CD-1 mice “… After screening of aggressor CD-1 mice, each CD-1 resident mouse received a wild-type intruder mouse and the two animals were separated by a perforated partition”

Response to the Comment # 11

We are very sorry for lack of the information about mice housing. During the 10 days of SS, resident mouse and CD-1 mouse were housed in the same cage with a perforated partition. After behavior analysis, resident mice were housed in groups (3~5 per cage). We revised our manuscript in the Materials and Methods section stated as below in line 81 2nd page.

After behavior analysis, mice were housed in groups (3~5 per cage) and fed an HFD (energy content: 62% fat, 18.2% protein, and 19.6% carbohydrate; Oriental Yeast Co., Tokyo, Japan) for 6 weeks.

Reviewer’s Comment # 12

Related to this, to make data more representative, photographs in Panel 2A should include lower magnification fields showing the presence of not only one but if possible several crown-like structures, even when adipocyte size is decreased in the final picture. Same for pictures 3A and 4C.

Response to the Comment # 12

          To respond to the Reviewer’s important comment, we revised Fig. 2A. In the case of Fig. 3A and Fig. 4C, we first performed the overview by rapid scanning of CLS at low-power field and then took a picture at high-power field to prevent the decline of the fluorescence intensity.

Reviewer’s Comment # 13

In section “2.12. Statistical analysis:”… followed by a Tukey-Kramer test to analyze significant differences between the groups. A p-value above05 was considered statistically significant.” Do you mean BELOW 0.05? Is this test different form other p-value based analysis or is this a typo?

Response to the Comment # 13

We corrected this typographic error in line 214 page 5 in the Revised Materials and Methods section. We would like to thank you very much for your careful attention.

Reviewer’s Comment # 14

Please specify how precisely cumulative caloric intake was measured for Suppl Fig 2C

Response to the Comment # 14

          According to the Reviewer’s important comment, we stated in 95 page 3 in the Materials and Methods section as below.

Cumulative caloric intake over 6 weeks was calculated by multiplying the total weight of food intake by total calorie of food pellets (5.062 kcal/g).

Reviewer’s Comment # 15

It would be advisable to discuss the conclusions drawn from differences observed between the increased ex vivo NE activity in eWAT while no differences were found in liver. Since the latter organ is key for the maintenance of glucose homeostasis and IR.

Response to the Comment # 15

To respond to the Reviewer’s important comment, we also examined the effect of SS on the NE activity in the lower limbs in which skeletal muscle was involved. However, there was no difference in NE activity between the two groups (Supplementary Fig. S7).

Elgazar-Carmon V et al. reported that neutrophils transiently infiltrate the parenchyma of intra-abdominal adipose tissue of HFD-fed mice and also observed that a physical binding between neutrophils and adipocytes using immunohistochemistry analysis (Ref.33 in the text, see below). Consistent with in vivo findings, they demonstrated that adherence of mouse peritoneal neutrophils to a monolayer of 3T3-L1 mouse adipocytes was significantly exaggerated after stimulation of neutrophils upon IL-8 analogue CXCL1 chemokine. They also showed that the adherence of neutrophils was prevented by preincubating with anti-ICAM-1 antibodies. Furthermore, Brake DK et al. reported that ICAM-1 mRNA expression was significantly increased in the abdominal fat of HFD-fed mice in a tissue-specific manner (Ref.34 in the text, see below). Considering that ICAM-1 expression in eWAT was comparable between stressed and control mice (Supplementary Fig. S7), and that CXCL2/MIP2 (murine IL-8 homologue) expression was significantly elevated in peripheral blood neutrophils of stressed mice, SS-induced priming of neutrophils is likely to contribute to the eWAT-specific enhancement of NE activation in stressed mice.

Reference

  1. Elgazar-Carmon, V.; Rudich, A.; Hadad, N.; Levy, R. Neutrophils transiently infiltrate intra-abdominal fat early in the course of high-fat feeding. J Lipid Res 2008, 49, 1894-1903, doi: 10.1194/jlr.M800132-JLR200.
  2. Brake, D.K.; Smith, E.O.; Mersmann, H.; Smith, C.W.; Robker, R.L. ICAM-1 expression in adipose tissue: effects of diet-induced obesity in mice. Am J Physiol Cell Physiol 2006, 291, C1232-1239.

These were stated in line 298 page 8 in the Revised Results section and in line  439 page 13 in the Revised Discussion section.

Reviewer’s Comment # 16

Please check English throughout the MS to correct and avoid frequent errors such as:

-“… Blood glucose concentrations were measured at 0 min before and 15, 30, 60, 90, and 120 min after…”

-“… HSP72 were estimated using an ELISA kits (BA E-5200; Labor Diagnostika Nord, Nordhorn,  Germany; ab108821; Abcam plc, Cambridge, UK;…)”.

-“…2.11. In vitro activation of bone marrow (BM) neutrophils on formyl peptide receptor (FPR) 1”

-“…The 680 FAST comprises of two near-infrared 176 (NIR) fluorochromes…”

-“…noradrenalin­­_“ (pp6 Figure 1 legend)

Response to the Comment # 16

          Thank you for your careful attention. According to your comments, we corrected the sentence as below.

-“… Blood glucose concentrations were measured prior to and 15, 30, 60, 90, and 120 min after…” (in line 110 page 3)

-“… HSP72 were estimated using an ELISA kits (BA E-5200; Labor Diagnostika Nord, Nordhorn, Germany; ab108821; Abcam plc, Cambridge, UK;…)”. (in line 119 page 3)

-“…2.11. In vitro activation of bone marrow (BM) neutrophils upon stimulation with formyl peptide receptor (FPR) 1 agonist” (in line 198 page 5)

-“…The 680 FAST comprises of two near-infrared 176 (NIR) fluorochromes…” (in line 187 page 4)

-“…noradrenaline­­_“ (pp6 Figure 1 legend) (in 248 page 6)

Reviewer’s Comment # 17

-Clarify what is meant by “these findings are not likely the case with the pathogenesis of IR in the clinical settings of metabolic syndrome [5,6]. “ (pp12, line 345).

Response to the Comment # 17

          To respond to the Reviewer’s important comment, we modified the sentence as below (in line 384 page 12). These findings are not likely the case with the pathogenesis of IR in the clinical setting because BW and adipocyte size in metabolic syndrome usually increase, accompanied by the increase in adipose tissue weight (Ref. 5,6 in the text, see below).

Reference

  1. Guilherme, A.; Virbasius, J.V.; Puri, V.; Czech, M.P. Adipocyte dysfunctions linking obesity to insulin resistance and type 2 diabetes. Nat Rev Mol Cell Biol 2008, 9, 367-377, doi:10.1038/nrm2391.
  2. Odegaard, J.I.; Chawla, A. Pleiotropic actions of insulin resistance and inflammation in metabolic homeostasis. Science 2013, 339, 172-177, doi:10.1126/science.1230721.

All authors are very grateful to Reviewer 3 for carefully reviewed comments and suggestions. We strongly hope that this revised version will be acceptable by the Reviewer 3.

Round 2

Reviewer 2 Report

I'm happy with this revised version of the manuscript and I think especially the methods section has clearly improved by the clarifying descriptions and supplemental figures 1 and 2. The answers to my questions were also satisfying and I recommend this paper to be accepted for publication in Cells.

Author Response

All authors are very grateful to Reviewer 2 for carefully reviewed comments and suggestions.

Reviewer 3 Report

Overall, my comments and suggestions have been addressed. However, a much more CAREFUL English editing needs to be performed since some mistakes as those specified below can still be found (only as examples, there are other throughout the text):

pp5, line 232 “stress levels, this finding needS to be interpreted with caution. “

pp14, line 480 “ … genetic deletion of NE in mice exerted more insulin sensitive than wild-type mice …”

Author Response

Response to the Reviewer 3

Thank you very much for your careful reading of our manuscript and helpful comments. According to your comments, we corrected some grammatical mistakes and added the missing information of qPCR primers. Additionally, we modified the sentence as you previously pointed out. We hope that this revised manuscript is improved well by your helpful comments.

Reviewer’s Comment

Overall, my comments and suggestions have been addressed. However, a much more CAREFUL English editing needs to be performed since some mistakes as those specified below can still be found (only as examples, there are other throughout the text):

1) pp5, line 232 “stress levels, this finding needS to be interpreted with caution. “

2) pp14, line 480 “ … genetic deletion of NE in mice exerted more insulin sensitive than wild-type mice …”

Response to the Comment # 1

We are very sorry for bothering you and we really appreciate your careful reading. According to your comments, we corrected the grammatical mistakes as below including other through the text.

  1. pp5, line 237 “… their stress levels, this finding needs to be interpreted with caution.”

  1. pp14, line 485 “As well as the pharmacological inhibition of NE, mice genetically deficient in NE exhibited more insulin sensitivity than wild-type mice after HFD feeding [16].”

  1. pp2, line 77 “Thus, it was unlikely to share the food pellets and feces between compartments inside the same cage (Supplementary Fig. S1).”

  1. pp3, line 113 “…concentrations were measured prior to and 30, 60, 90, and 120 min after injection.”

  1. pp3, line 136 “…were assessed per 3 sections from 8-10 animals from each group.”

  1. pp3, line 138 “The total number of 4 to 6 representative images was chosen from each animals at random, and then analyzed.”

  1. pp5, line 197 “…the liver, eWAT, and lower limbs were harvested…”

  1. pp5, line 229 “Glucose tolerance was significantly impaired in HFD-fed mice compared with that in mice before HFD feeding…”

  1. pp5, line 231 “…resulting in the significantly higher AUC than that in HFD-fed control mice ( 1C).”

  1. pp8, line 299 “…the extent of which was much higher than that in HFD-fed control mice…”

  1. pp13, line 407 “…on the late phase of IR development needs to be investigated in future studies.”

  1. pp14, line 446 “…neutrophils transiently infiltrated the parenchyma…”

  1. pp14, line 447 “…and also observed that a physical binding between…”

  1. pp14, line 460 “Walsh DE et al. reported that activated NE functioned as a Toll-like receptor (TLR) 4 activator…”

  1. pp14, line 462 “…stimulation of intraperitoneal macrophages upon recombinant mouse NE significantly increased TNF-α mRNA expression…”

  1. pp14, line 468 “Accumulation of ATMs and subsequent inflammatory response…”

  1. pp14, line 475 “…stress-associated whole-body IR needs to be investigated in future studies.”

  1. pp14, line 483 “…IR development and shed new insights into the underlying mechanism…”

We also added the missing information of qPCR primers in line 156 page 4 in the Materials and Methods section.

MCP-1: forward, 5’-GGCTCAGCCAGATGCAGTTAA-3’; reverse, 5’- CCTACTCATTGGGATCATCTTGCT-3’; ICAM-1: forward, 5’-AGCACCTCCCCACCTACTTT-3’; reverse, 5’-AGCTTGCACGACCCTTCTAA-3’; IL-1β: forward, 5’-AGAGCCCATCCTCTGTGACTCA-3’; reverse, 5’-TCATATGGGTCCGACAGCACGA-3’; IL-6: forward, 5’-ACAACCACGGCCTTCCCTACTT-3’; reverse, 5’-CACGATTTCCCAGAGAACATGTG-3’;

In addition, according to the reviewer’ previous comment, we corrected the sentence in line 59 page 2 as below.

“In the current study, we show that socially stressed mice are susceptible to the development of HFD-induced IR,…”

All authors are very grateful to Reviewer 3 for carefully reviewed comments and suggestions. We strongly hope that this revised version will be acceptable by the Reviewer 3.